# Two Methods for Wild Variational Inference

**Qiang Liu**    **Yihao Feng**
Computer Science, Dartmouth College
Hanover, NH, 03755
{qiang.liu, yihao.feng.gr}@dartmouth.edu

## Abstract

Variational inference provides a powerful tool for approximate probabilistic inference on complex, structured models. Typical variational inference methods, however, require to use inference networks with computationally tractable probability density functions. This largely limits the design and implementation of variational inference methods. We consider *wild variational inference* methods that do not require tractable density functions on the inference networks, and hence can be applied in more challenging cases. As an example of application, we treat stochastic gradient Langevin dynamics (SGLD) as an inference network, and use our methods to automatically adjust the step sizes of SGLD, yielding significant improvement over the hand-designed step size schemes.

## 1 Introduction

Probabilistic modeling provides a principled approach for reasoning under uncertainty, and has been increasingly dominant in modern machine learning where highly complex, structured probabilistic models are often the essential components for solving complex problems with increasingly larger datasets. A key challenge, however, is to develop computationally efficient Bayesian inference methods to approximate, or draw samples from the posterior distributions. Variational inference (VI) provides a powerful tool for scaling Bayesian inference to complex models and big data. The basic idea of VI is to approximate the true distribution with a simpler distribution by minimizing the KL divergence, transforming the inference problem into an optimization problem, which is often then solved efficiently using stochastic optimization techniques (e.g., Hoffman et al., 2013; Kingma & Welling, 2013). However, the practical design and application of VI are still largely restricted by the requirement of using simple approximation families, as we explain in the sequel.

Let $p(z)$ be a distribution of interest, such as the posterior distribution in Bayesian inference. VI approximates $p(z)$ with a simpler distribution $q^*(z)$ found in a set $\mathcal{Q} = \{q_\eta(z)\}$ of distributions indexed by parameter $\eta$ by minimizing the KL divergence objective:

$$\min_\eta \left\{ \mathrm{KL}(q_\eta \,||\, p) \equiv \mathbb{E}_{z \sim q_\eta}[\log(q_\eta(z)/p(z))] \right\}, \tag{1}$$

where we can get exact result $p = q^*$ if $\mathcal{Q}$ is chosen to be broad enough to actually include $p$. In practice, however, $\mathcal{Q}$ should be chosen carefully to make the optimization in (1) computationally tractable; this casts two constraints on $\mathcal{Q}$:

1. A minimum requirement is that we should be able to sample from $q_\eta$ efficiently, which allows us to make estimates and predictions based on $q_\eta$ in placement of the more intractable $p$. The samples from $q_\eta$ can also be used to approximate the expectation $\mathbb{E}_q[\cdot]$ in (1) during optimization. This means that there should exist some computable function $f(\eta; \xi)$, called the *inference network*, which takes a random seed $\xi$, whose distribution is denoted by $q_0$, and outputs a random variable $z = f(\eta; \xi)$ whose distribution is $q_\eta$.

2. We should also be able to calculate the density $q_\eta(z)$ or it is derivative in order to optimize the KL divergence in (1). This, however, casts a much more restrictive condition, since it requires us to use only simple inference network $f(\eta; \xi)$ and input distributions $q_0$ to ensure a tractable form for the density $q_\eta$ of the output $z = f(\eta; \xi)$.

In fact, it is this requirement of calculating $q_\eta(z)$ that has been the major constraint for the design of state-of-the-art variational inference methods. The traditional VI methods are often limited to

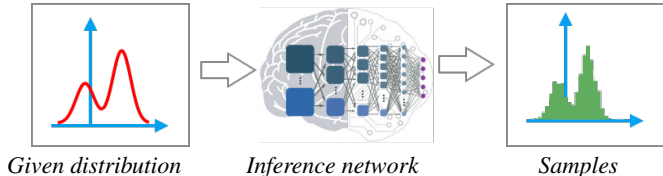

*Given distribution*　　　*Inference network*　　　*Samples*

Figure 1: Wild variational inference allows us to train general stochastic neural inference networks to learn to draw (approximate) samples from the target distributions, without restriction on the computational tractability of the density function of the neural inference networks.

using simple mean field, or Gaussian-based distributions as $q_\eta$ and do not perform well for approximating complex target distributions. There is a line of recent work on variational inference with rich approximation families (e.g., Rezende & Mohamed, 2015b; Tran et al., 2015; Ranganath et al., 2015, to name only a few), all based on handcrafting special inference networks to ensure the computational tractability of $q_\eta(z)$ while simultaneously obtaining high approximation accuracy. These approaches require substantial mathematical insights and research effects, and can be difficult to understand or use for practitioners without a strong research background in VI. Methods that allow us to use arbitrary inference networks without substantial constraints can significantly simplify the design and applications of VI methods, allowing practical users to focus more on choosing proposals that work best with their specific tasks.

We use the term *wild variational inference* to refer to variants of variational methods working with general inference networks $f(\eta, \xi)$ without tractability constraints on its output density $q_\eta(z)$; this should be distinguished with the *black-box variational inference* (Ranganath et al., 2014) which refers to methods that work for generic target distributions $p(z)$ without significant model-by-model consideration (but still require to calculate the proposal density $q_\eta(z)$). Essentially, wild variational inference makes it possible to "learn to draw samples", constructing black-box neural samplers for given distributions. This enables more adaptive and automatic design of efficient Bayesian inference procedures, replacing the hand-designed inference algorithms with more efficient ones that can improve their efficiency adaptively over time based on past tasks they performed.

In this work, we discuss two methods for wild variational inference, both based on recent works that combine kernel techniques with Stein's method (e.g., Liu & Wang, 2016; Liu et al., 2016). The first method, also discussed in Wang & Liu (2016), is based on iteratively adjusting parameter $\eta$ to make the random output $z = f(\eta; \xi)$ mimic a Stein variational gradient direction (SVGD) (Liu & Wang, 2016) that optimally decreases its KL divergence with the target distribution. The second method is based on minimizing a kernelized Stein discrepancy, which, unlike KL divergence, does not require to calculate density $q_\eta(z)$ for the optimization thanks to its special form.

Another critical problem is to design good network architectures well suited for Bayesian inference. Ideally, the network design should leverage the information of the target distribution $p(z)$ in a convenient way. One useful perspective is that we can view the existing MC/MCMC methods as (hand-designed) stochastic neural networks which can be used to construct *native* inference networks for given target distributions. On the other hand, using existing MC/MCMC methods as inference networks also allow us to adaptively adjust the hyper-parameters of these algorithms; this enables *amortized inference* which leverages the experience on past tasks to accelerate the Bayesian computation, providing a powerful approach for designing efficient algorithms in settings when a large number of similar tasks are needed.

As an example, we leverage stochastic gradient Langevin dynamics (SGLD) (Welling & Teh, 2011) as the inference network, which can be treated as a special deep residential network (He et al., 2016), in which important gradient information $\nabla_z \log p(z)$ is fed into each layer to allow efficient approximation for the target distribution $p(z)$. In our case, the network parameter $\eta$ are the step sizes of SGLD, and our method provides a way to adaptively improve the step sizes, providing speed-up on future tasks with similar structures. We show that the adaptively estimated step sizes significantly outperform the hand-designed schemes such as Adagrad.

**Related Works**　The idea of amortized inference (Gershman & Goodman, 2014) has been recently applied in various domains of probabilistic reasoning, including both amortized variational inference

(e.g., Kingma & Welling, 2013; Rezende & Mohamed, 2015a) and date-driven designs of Monte Carlo based methods (e.g., Paige & Wood, 2016), to name only a few. Most of these methods, however, require to explicitly calculate $q_\eta(z)$ (or its gradient).

One well exception is a very recent work (Ranganath et al., 2016) that also avoids calculating $q_\eta(z)$ and hence works for general inference networks; their method is based on a similar idea related to Stein discrepancy (Liu et al., 2016; Oates et al., 2017; Chwialkowski et al., 2016; Gorham & Mackey, 2015), for which we provide a more detailed discussion in Section 3.2.

The auxiliary variational inference methods (e.g., Agakov & Barber, 2004) provide an alternative way when the variational distribution $q_\eta(z)$ can be represented as a hidden variable model. In particular, Salimans et al. (2015) used the auxiliary variational approach to leverage MCMC as a variational approximation. These approaches, however, still require to write down the likelihood function on the augmented spaces, and need to introduce an additional inference network related to the auxiliary variables.

There is a large literature on traditional adaptive MCMC methods (e.g., Andrieu & Thoms, 2008; Roberts & Rosenthal, 2009) which can be used to adaptively adjust the proposal distribution of MCMC by exploiting the special theoretical properties of MCMC (e.g., by minimizing the auto-correlation). Our method is simpler, more generic, and works efficiently in practice thanks to the use of gradient-based back-propagation. Finally, connections between stochastic gradient descent and variational inference have been discussed and exploited in Mandt et al. (2016); Maclaurin et al. (2015).

**Outline**   Section 2 introduces background on Stein discrepancy and Stein variational gradient descent. Section 3 discusses two methods for wild variational inference. Section 4 discuss using stochastic gradient Langevin dynamics (SGLD) as the inference network. Empirical results are shown in Section 5.

## 2   STEIN'S IDENTITY, STEIN DISCREPANCY, STEIN VARIATIONAL GRADIENT

**Stein's identity**   Stein's identity plays a fundamental role in our framework. Let $p(z)$ be a positive differentiable density on $\mathbb{R}^d$, and $\boldsymbol{\phi}(z) = [\phi_1(z), \cdots, \phi_d(z)]^\top$ is a differentiable vector-valued function. Define $\nabla_z \cdot \boldsymbol{\phi} = \sum_i \partial_{z_i} \boldsymbol{\phi}$. Stein's identity is

$$\mathbb{E}_{z \sim p}[\langle \nabla_z \log p(z), \ \boldsymbol{\phi}(z) \rangle + \nabla_z \cdot \boldsymbol{\phi}(z)] = \int_{\mathcal{X}} \nabla_z \cdot (p(z)\boldsymbol{\phi}(z)) dx = 0, \tag{2}$$

which holds once $p(z)\boldsymbol{\phi}(z)$ vanishes on the boundary of $\mathcal{X}$ by integration by parts or Stokes' theorem; It is useful to rewrite Stein's identity in a more compact way:

$$\mathbb{E}_{z \sim p}[\mathcal{T}_p \boldsymbol{\phi}(z)] = 0, \quad \text{with} \quad \mathcal{T}_p \boldsymbol{\phi} \stackrel{def}{=} \langle \nabla_z \log p, \ \boldsymbol{\phi} \rangle + \nabla_z \cdot \boldsymbol{\phi}, \tag{3}$$

where $\mathcal{T}_p$ is called a *Stein operator*, which acts on function $\boldsymbol{\phi}$ and returns a zero-mean function $\mathcal{T}_p \boldsymbol{\phi}(z)$ under $z \sim p$. A key computational advantage of Stein's identity and Stein operator is that they depend on $p$ only through the derivative of the log-density $\nabla_z \log p(z)$, which does not depend on the cumbersome normalization constant of $p$, that is, when $p(z) = \bar{p}(z)/Z$, we have $\nabla_z \log p(z) = \nabla_z \log \bar{p}(z)$, independent of the normalization constant $Z$. This property makes Stein's identity a powerful practical tool for handling unnormalized distributions widely appeared in machine learning and statistics.

**Stein Discrepancy**   Although Stein's identity ensures that $\mathcal{T}_p \boldsymbol{\phi}$ has zero expectation under $p$, its expectation is generally non-zero under a different distribution $q$. Instead, for $p \neq q$, there must exist a $\boldsymbol{\phi}$ which distinguishes $p$ and $q$ in the sense that $\mathbb{E}_{z \sim q}[\mathcal{T}_p \boldsymbol{\phi}(z)] \neq 0$. Stein discrepancy leverages this fact to measure the difference between $p$ and $q$ by considering the "maximum violation of Stein's identity" for $\boldsymbol{\phi}$ in certain function set $\mathcal{F}$:

$$\mathbb{D}(q \,||\, p) = \max_{\boldsymbol{\phi} \in \mathcal{F}} \left\{ \mathbb{E}_{z \sim q}[\mathcal{T}_p \boldsymbol{\phi}(z)] \right\}, \tag{4}$$

where $\mathcal{F}$ is the set of functions $\boldsymbol{\phi}$ that we optimize over, and decides both the discriminative power and computational tractability of Stein discrepancy. Kernelized Stein discrepancy (KSD) is a special

Stein discrepancy that takes $\mathcal{F}$ to be the unit ball of vector-valued reproducing kernel Hilbert spaces (RKHS), that is,

$$\mathcal{F} = \{\phi \in \mathcal{H}^d \colon ||\phi||_{\mathcal{H}^d} \leq 1\}, \tag{5}$$

where $\mathcal{H}$ is a real-valued RKHS with kernel $k(z, z')$. This choice of $\mathcal{F}$ makes it possible to get a closed form solution for the optimization in (4) (Liu et al., 2016; Chwialkowski et al., 2016; Oates et al., 2017):

$$\mathbb{D}(q \,||\, p) = \max_{\phi \in \mathcal{H}^d} \left\{ \mathbb{E}_{z \sim q}[\mathcal{T}_p \phi(z)], \quad s.t. \quad ||\phi||_{\mathcal{H}^d} \leq 1 \right\}, \tag{6}$$

$$= \sqrt{\mathbb{E}_{z, z' \sim q}[\kappa_p(z, z')]}, \tag{7}$$

where $\kappa_p(z, z')$ is a positive definite kernel obtained by applying Stein operator on $k(z, z')$ twice:

$$\kappa_p(z, z') = \mathcal{T}_p^{z'}(\mathcal{T}_p^z \otimes k(z, z')),$$
$$= \boldsymbol{s}_p(z)\boldsymbol{s}_p(z')k(z, z') + \boldsymbol{s}_p(z)\nabla_{z'}k(z, z') + \boldsymbol{s}_p(z')\nabla_z k(z, z') + \nabla_z \cdot (\nabla_{z'}k(z, z')), \tag{8}$$

where $\boldsymbol{s}_p(z) = \nabla_z \log p(z)$ and $\mathcal{T}_p^z$ and $\mathcal{T}_p^z$ denote the Stein operator when treating $k(z, z')$ as a function of $z$ and $z'$, respectively; here we defined $\mathcal{T}_p^z \otimes k(z, z') = \nabla_x \log p(x)k(z, z') + \nabla_x k(z, z')$ which returns a $d \times 1$ vector-valued function. It can be shown that $\mathbb{D}(q \,||\, p) = 0$ if and only if $q = p$ when $k(z, z')$ is strictly positive definite in a proper sense (Liu et al., 2016; Chwialkowski et al., 2016). $\mathbb{D}(q \,||\, p)$ can treated as a variant of maximum mean discrepancy equipped with kernel $\kappa_p(z, z')$ which depends on $p$ (which makes $\mathbb{D}(q \,||\, p)$ asymmetric on $q$ and $p$).

The form of KSD in (6) allows us to estimate the discrepancy between a set of sample $\{z_i\}$ (e.g., drawn from $q$) and a distribution $p$ specified by $\nabla_z \log p(z)$,

$$\hat{\mathbb{D}}_u^2(\{z_i\} \,||\, p) = \frac{1}{n(n-1)} \sum_{i \neq j} [\kappa_p(z_i, z_j)], \qquad \hat{\mathbb{D}}_v^2(\{z_i\} \,||\, p) = \frac{1}{n^2} \sum_{i,j} [\kappa_p(z_i, z_j)], \tag{9}$$

where $\hat{\mathbb{D}}_u^2(q \,||\, p)$ provides an unbiased estimator (hence called a $U$-statistic) for $\mathbb{D}^2(q \,||\, p)$, and $\hat{\mathbb{D}}_v^2(q \,||\, p)$, called $V$-statistic, provides a biased estimator but is guaranteed to be always non-negative: $\hat{\mathbb{D}}_v^2(\{z_i\} \,||\, p) \geq 0$.

**Stein Variational Gradient Descent (SVGD)**   Stein operator and Stein discrepancy have a close connection with KL divergence, which is exploited in Liu & Wang (2016) to provide a general purpose deterministic approximate sampling method. Assume that $\{z_i\}_{i=1}^n$ is a sample (or a set of particles) drawn from $q$, and we want to update $\{z_i\}_{i=1}^n$ to make it "move closer" to the target distribution $p$ to improve the approximation quality. We consider updates of form

$$z_i \leftarrow z_i + \epsilon \boldsymbol{\phi}^*(z_i), \quad \forall i = 1, \ldots, n, \tag{10}$$

where $\phi^*$ is a perturbation direction, or velocity field, chosen to maximumly decrease the KL divergence between the distribution of updated particles and the target distribution, in the sense that

$$\boldsymbol{\phi}^* = \arg\max_{\boldsymbol{\phi} \in \mathcal{F}} \left\{ -\frac{d}{d\epsilon} \mathrm{KL}(q_{[\epsilon\boldsymbol{\phi}]} \,||\, p)\big|_{\epsilon=0} \right\}, \tag{11}$$

where $q_{[\epsilon\phi]}$ denotes the density of the updated particle $z' = z + \epsilon\phi(z)$ when the density of the original particle $z$ is $q$, and $\mathcal{F}$ is the set of perturbation directions that we optimize over. A key observation (Liu & Wang, 2016) is that the optimization in (11) is in fact equivalent to the optimization for KSD in (4); we have

$$-\frac{d}{d\epsilon} \mathrm{KL}(q_{[\epsilon\phi]} \,||\, p)\big|_{\epsilon=0} = \mathbb{E}_{z \sim q}[\mathcal{T}_p \phi(z)], \tag{12}$$

that is, the Stein operator transforms the perturbation $\phi$ on the random variable (the particles) to the change of the KL divergence. Taking $\mathcal{F}$ to be unit ball of $\mathcal{H}^d$ as in (5), the optimal solution $\phi^*$ of (11) equals that of (6), which is shown to be (e.g., Liu et al., 2016)

$$\boldsymbol{\phi}^*(z') \propto \mathbb{E}_{z \sim q}[\mathcal{T}_p^z k(z, z')] = \mathbb{E}_{z \sim q}[\nabla_z \log p(z)k(z, z') + \nabla_z k(z, z')].$$

---

**Algorithm 1** Amortized SVGD and KSD Minimization for Wild Variational Inference

---

**for** iteration $t$ **do**

 1. Draw random $\{\xi_i\}_{i=1}^n$, calculate $z_i = f(\eta; \xi_i)$, and the Stein variational gradient $\Delta z_i$ in (13).

 2. Update parameter $\eta$ using (14) or (15) for amortized SVGD, or (17) for KSD minimization.

**end for**

---

By approximating the expectation under $q$ with the empirical mean of the current particles $\{z_i\}_{i=1}^n$, SVGD admits a simple form of update that iteratively moves the particles towards the target distribution,

$$z_i \leftarrow z_i + \epsilon \Delta z_i, \quad \forall i = 1, \ldots, n,$$
$$\Delta z_i = \hat{\mathbb{E}}_{z \in \{z_i\}_{i=1}^n}[\nabla_z \log p(z)k(z, z_i) + \nabla_z k(z, z_i)], \tag{13}$$

where $\hat{\mathbb{E}}_{z \sim \{z_i\}_{i=1}^n}[f(z)] = \sum_i f(z_i)/n$. The two terms in $\Delta z_i$ play two different roles: the term with the gradient $\nabla_z \log p(z)$ drives the particles towards the high probability regions of $p(z)$, while the term with $\nabla_z k(z, z_i)$ serves as a repulsive force to encourage diversity; to see this, consider a stationary kernel $k(z, z') = k(z - z')$, then the second term reduces to $\hat{\mathbb{E}}_z \nabla_z k(z, z_i) = -\hat{\mathbb{E}}_z \nabla_{z_i} k(z, z_i)$, which can be treated as the negative gradient for minimizing the average similarity $\hat{\mathbb{E}}_z k(z, z_i)$ in terms of $z_i$.

It is easy to see from (13) that $\Delta z_i$ reduces to the typical gradient $\nabla_z \log p(z_i)$ when there is only a single particle ($n = 1$) and $\nabla_z k(z, z_i)$ when $z = z_i$, in which case SVGD reduces to the standard gradient ascent for maximizing $\log p(z)$ (i.e., maximum *a posteriori* (MAP)).

## 3    TWO METHODS FOR WILD VARIATIONAL INFERENCE

Since the direct parametric optimization of the KL divergence (1) requires calculating $q_\eta(z)$, there are two essential ways to avoid calculating $q_\eta(z)$: either using alternative (approximate) optimization approaches, or using different divergence objective functions. We discuss two possible approaches in this work: one based on "amortizing SVGD" (Wang & Liu, 2016) which trains the inference network $f(\eta, \xi)$ so that its output mimic the SVGD dynamics in order to decrease the KL divergence; another based on minimizing the KSD objective (9) which does not require to evaluate $q(z)$ thanks to its special form.

### 3.1    AMORTIZED SVGD

SVGD provides an optimal updating direction to iteratively move a set of particles $\{z_i\}$ towards the target distribution $p(z)$. We can leverage it to train an inference network $f(\eta; \xi)$ by iteratively adjusting $\eta$ so that the output of $f(\eta; \xi)$ changes along the Stein variational gradient direction in order to maximumly decrease its KL divergence with the target distribution. By doing this, we "amortize" SVGD into a neural network, which allows us to leverage the past experience to adaptively improve the computational efficiency and generalize to new tasks with similar structures. Amortized SVGD is also presented in Wang & Liu (2016); here we present some additional discussion.

To be specific, assume $\{\xi_i\}$ are drawn from $q_0$ and $z_i = f(\eta; \xi_i)$ the corresponding random output based on the current estimation of $\eta$. We want to adjust $\eta$ so that $z_i$ changes along the Stein variational gradient direction $\Delta z_i$ in (13) so as to maximumly decrease the KL divergence with target distribution. This can be done by updating $\eta$ via

$$\eta \leftarrow \arg\min_\eta \sum_{i=1}^n ||f(\eta; \xi_i) - z_i - \epsilon \Delta z_i||_2^2. \tag{14}$$

Essentially, this projects the non-parametric perturbation direction $\Delta z_i$ to the change of the finite dimensional network parameter $\eta$. If we take the step size $\epsilon$ to be small, then the updated $\eta$ by (14) should be very close to the old value, and a single step of gradient descent of (14) can provide a

good approximation for (14). This gives a simpler update rule:

$$\eta \leftarrow \eta + \epsilon \sum_i \partial_\eta f(\eta; \, \xi_i) \Delta z_i, \tag{15}$$

which can be intuitively interpreted as a form of chain rule that *back-propagates the SVGD gradient to the network parameter $\eta$*. In fact, when we have only one particle, (15) reduces to the standard gradient ascent for $\max_\eta \log p(f(\eta; \, \xi))$, in which $f_\eta$ is trained to "learn to optimize" (e.g., Andrychowicz et al., 2016), instead of "learn to sample" $p(z)$. Importantly, as we have more than one particles, the repulsive term $\nabla_z k(z, z_i)$ in $\Delta z_i$ becomes active, and enforces an amount of diversity on the network output that is consistent with the variation in $p(z)$. The full algorithm is summarized in Algorithm 1.

Amortized SVGD can be treated as minimizing the KL divergence using a rather special algorithm: it leverages the non-parametric SVGD which can be treated as approximately solving the infinite dimensional optimization $\min_q \mathrm{KL}(q \, || \, p)$ without explicitly assuming a parametric form on $q$, and iteratively *projecting* the non-parametric update back to the finite dimensional parameter space of $\eta$. It is an interesting direction to extend this idea to "amortize" other MC/MCMC-based inference algorithms. For example, given a MCMC with transition probability $T(z'|z)$ whose stationary distribution is $p(z)$, we may adjust $\eta$ to make the network output move towards the updated values $z'$ drawn from the transition probability $T(z'|z)$. The advantage of using SVGD is that it provides a *deterministic* gradient direction which we can *back-propagate* conveniently and is particle efficient in that it reduces to "learning to optimize" with a single particle. We have been using the simple $L^2$ loss in (14) mainly for convenience; it is possible to use other two-sample discrepancy measures such as maximum mean discrepancy.

## 3.2 KSD Variational Inference

Amortized SVGD attends to minimize the KL divergence objective, but can not be interpreted as a typical finite dimensional optimization on parameter $\eta$. Here we provide an alternative method based on directly minimizing the kernelized Stein discrepancy (KSD) objective, for which, thanks to its special form, the typical gradient-based optimization can be performed without needing to estimate $q(z)$ explicitly.

To be specific, take $q_\eta$ to be the density of the random output $z = f(\eta; \, \xi)$ when $\xi \sim q_0$, and we want to find $\eta$ to minimize $\mathbb{D}(q_\eta \, || \, p)$. Assuming $\{\xi_i\}$ is i.i.d. drawn from $q_0$, we can approximate $\mathbb{D}^2(q_\eta \, || \, p)$ unbiasedly with a U-statistics:

$$\mathbb{D}^2(q_\eta \, || \, p) \approx \frac{1}{n(n-1)} \sum_{i \neq j} \kappa_p(f(\eta; \, \xi_i), \, f(\eta; \, \xi_j)), \tag{16}$$

for which a standard gradient descent can be derived for optimizing $\eta$:

$$\eta \leftarrow \eta - \epsilon \frac{2}{n(n-1)} \sum_{i \neq j} \partial_\eta f(\eta; \, \xi_i) \nabla_{z_i} \kappa_p(z_i, z_j), \quad \text{where} \quad z_i = f(\eta; \, \xi_i). \tag{17}$$

This enables a wild variational inference method based on directly minimizing $\eta$ with standard (stochastic) gradient descent. See Algorithm 1. Note that (17) is similar to (15) in form, but replaces $\Delta z_i$ with a $\tilde{\Delta} z_i \propto - \sum_{j: \, i \neq j} \nabla_{z_i} \kappa_p(z_i, z_j)$. It is also possible to use the $V$-statistic in (9), but we find that the $U$-statistic performs much better in practice, possibly because of its unbiasedness property.

Minimizing KSD can be viewed as minimizing a constrastive divergence objective function. To see this, recall that $q_{[\epsilon \phi]}$ denotes the density of $z' = z + \epsilon \phi(z)$ when $z \sim q$. Combining (11) and (6), we can show that

$$\mathbb{D}^2(q \, || \, p) \approx \frac{1}{\epsilon}(\mathrm{KL}(q \, || \, p) - \mathrm{KL}(q_{[\epsilon \phi]} \, || \, p)).$$

That is, KSD measures the amount of decrease of KL divergence when we update the particles along the optimal SVGD perturbation direction $\phi$ given by (11). If $q = p$, then the decrease of KL

divergence equals zero and $\mathbb{D}^2(q \,||\, p)$ equals zero. In fact, as shown in Liu & Wang (2016) KSD can be explicitly represented as the magnitude of a functional gradient of KL divergence:

$$\mathbb{D}(q \,||\, p) = \left\| \frac{d}{d\boldsymbol{\phi}} \mathrm{KL}(q_{[\boldsymbol{\phi}]} \,||\, p) \big|_{\boldsymbol{\phi}=0} \right\|_{\mathcal{H}^d},$$

where $q_{[\boldsymbol{\phi}]}$ is the density of $z = z + \boldsymbol{\phi}(z)$ when $z \sim q$, and $\frac{d}{d\boldsymbol{\phi}} F(\boldsymbol{\phi})$ denotes the functional gradient of functional $F(\boldsymbol{\phi})$ w.r.t. $\boldsymbol{\phi}$ defined in RKHS $\mathcal{H}^d$, and $\frac{d}{d\boldsymbol{\phi}} F(\boldsymbol{\phi})$ is also an element in $\mathcal{H}^d$. Therefore, KSD variational inference can be treated as explicitly minimizing the magnitude of the gradient of KL divergence, in contract with amortized SVGD which attends to minimize the KL divergence objective itself.

This idea is also similar to the contrastive divergence used for learning restricted Boltzmann machine (RBM) (Hinton, 2002) (which, however, optimizes $p$ with fixed $q$). It is possible to extend this approach by replacing $z' = z + \epsilon \boldsymbol{\phi}(z)$ with other transforms, such as these given by a transition probability of a Markov chain whose stationary distribution is $p$. In fact, according the so called *generator method* for constructing Stein operator (Barbour, 1988), any generator of a Markov process defines a Stein operator that can be used to define a corresponding Stein discrepancy.

This idea is related to a very recent work by Ranganath et al. (2016), which is based on directly minimizing the variational form of Stein discrepancy in (4); Ranganath et al. (2016) assumes $\mathcal{F}$ consists of a neural network $\boldsymbol{\phi}_\tau(z)$ parametrized by $\tau$, and find $\eta$ by solving the following min-max problem:

$$\min_{\eta} \max_{\tau} \mathbb{E}_{z \sim q}[\mathcal{T}_p \boldsymbol{\phi}_\tau(z)].$$

In contrast, our method leverages the closed form solution by taking $\mathcal{F}$ to be an RKHS and hence obtains an explicit optimization problem, instead of a min-max problem that can be computationally more expensive, or have difficulty in achieving convergence.

Because $\kappa_p(x, x')$ (defined in (8)) depends on the derivative $\nabla_x \log p(x)$ of the target distribution, the gradient in (17) depends on the Hessian matrix $\nabla_x^2 \log p(x)$ and is hence less convenient to implement compared with amortized SVGD (the method by Ranganath et al. (2016) also has the same problem). However, this problem can be alleviated using automatic differentiation tools, which be used to directly take the derivative of the objective in (16) without manually deriving its derivatives.

## 4 Langevin Inference Network

With wild variational inference, we can choose more complex inference network structures to obtain better approximation accuracy. Ideally, the best network structure should leverage the special properties of the target distribution $p(z)$ in a convenient way. One way to achieve this by viewing existing MC/MCMC methods as inference networks with hand-designed (and hence potentially suboptimal) parameters, but good architectures that take the information of the target distribution $p(z)$ into account. By applying wild variational inference on networks constructed based on existing MCMC methods, we effectively provide an hyper-parameter optimization for these existing methods. This allows us to fully optimize the potential of existing Bayesian inference methods, significantly improving the result with less computation cost, and decreasing the need for hyper-parameter tuning by human experts. This is particularly useful when we need to solve a large number of similar tasks, where the computation cost spent on optimizing the hyper-parameters can significantly improve the performance on the future tasks.

**Stochastic Gradient Langevin Dynamics**   We first take the original stochastic gradient Langevin dynamics (SGLD) algorithm (Welling & Teh, 2011) as an example. SGLD starts with a random initialization $z_0$, and perform iterative update of form

$$z^{t+1} \leftarrow z^t + \eta^t \odot \nabla_z \log \hat{p}(z^t; \mathbb{M}^t) + \sqrt{2\eta^t} \odot \xi^t, \quad \forall t = 1, \cdots T, \tag{18}$$

where $\log \hat{p}(z^t; \mathbb{M}^t)$ denotes an approximation of $\log p(z^t)$ based on, e.g., a random mini-batch $\mathbb{M}^t$ of observed data at $t$-th iteration, and $\xi^t$ is a standard Gaussian random vector of the same size as $z$, and $\eta^t$ denotes a (vector) step-size at $t$-th iteration; here "$\odot$" denotes element-wise product. When running SGLD for $T$ iterations, we can treat $z^T$ as the output of a $T$-layer neural network

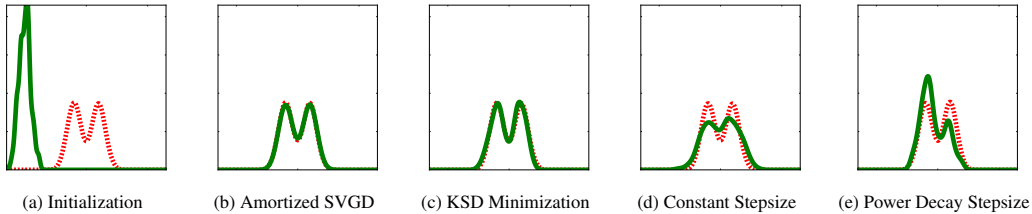

| (a) Initialization | (b) Amortized SVGD | (c) KSD Minimization | (d) Constant Stepsize | (e) Power Decay Stepsize |

Figure 2: Results on a 1D Gaussian mixture when training the step sizes of SGLD with $T = 20$ iterations. The target distribution $p(x)$ is shown by the red dashed line. (a) The distribution of the initialization $z_0$ of SGLD (the green line), visualized by kernel density estimator. (b)-(d) The distribution of the final output $z^T$ (green line) given by different types of step sizes, visualized by kernel density estimator.

parametrized by the collection of step sizes $\eta = \{\eta^t\}_{t=1}^T$, whose random inputs include the random initialization $z_0$, the mini-batch $\mathbb{M}^t$ and Gaussian noise $\xi^t$ at each iteration $t$. We can see that this defines a rather complex network structure with several different types of random inputs ($z^0$, $\mathbb{M}^t$ and $\xi^t$). This makes it intractable to explicitly calculate the density of $z^T$ and traditional variational inference methods can not be applied directly. But wild variational inference can still allow us to adaptively improve the optimal step-size $\eta$ in this case.

**General Langevin Networks**   Based on the original formula of SGLD, we proposed a more general langevin network structure, and each layer of the network has a form

$$z^{t+1} \leftarrow A^t z^t + h(B^t B^{t^\top} \nabla_z \log \hat{p}(z^t; \mathbb{M}^t) + B^t \xi^t + D^t), \qquad \forall t = 1, \cdots T, \qquad (19)$$

where $A^t$, $B^t$ and $D^t$ are network parameters at $t$-th iteration(whose size is $d \times d$, and $d$ is the size of $z^t$), and $h(\cdot)$ denotes a smooth element-wise non-linearity function; here $\xi^t$ is still a standard gaussian random vector with the same size as $z$. With this more complex network, we can use fewer layers to construct more powerful back-box samplers.

## 5   EMPIRICAL RESULTS

### 5.1   SGLD INFERENCE NETWORK

We first test our algorithm with SGLD inference network with (18) formula on both a toy Gaussian mixture model and a Bayesian logistic regression example. We find that we can adaptively learn step sizes that significantly outperform the existing hand-designed step size schemes, and hence save computational cost in the testing phase. In particular, we compare with the following step size schemes, for all of which we report the best results (testing accuracy in Figure 3(a); testing likelihood in Figure 3(b)) among a range of hyper-parameters:

1. *Constant Step Size*. We select a best constant step size in $\{1, 2, 2^3, \ldots, 2^{29}\} \times 10^{-6}$.

2. *Power Decay Step Size*. We consider $\epsilon^t = 10^a \times (b + t)^{-\gamma}$ where $\gamma = 0.55$, $a \in \{-6, -5, \ldots, 1, 2\}$, $b \in \{0, 1, \ldots, 9\}$.

3. *Adagrad*, *Rmsprop*, *Adadelta*, all with the master step size selected in $\{1, 2, 2^3, \ldots, 2^{29}\} \times 10^{-6}$, with the other parameters chosen by default values.

**Gaussian Mixture**   We start with a simple 1D Gaussian mixture example shown in Figure 2 where the target distribution $p(z)$ is shown by the red dashed curve. We use amortized SVGD and KSD to optimize the step size parameter of the Langevin inference network in (18) with $T = 20$ layers (i.e., SGLD with $T = 20$ iterations), with an initial $z_0$ drawn from a $q_0$ far away from the target distribution (see the green curve in Figure 2(a)); this makes it critical to choose a proper step size to achieve close approximation within $T = 20$ iterations. We find that amortized SVGD and KSD allow us to achieve good performance with 20 steps of SGLD updates (Figure 2(b)-(c)), while the result of the best constant step size and power decay step-size are much worse (Figure 2(d)-(e)).

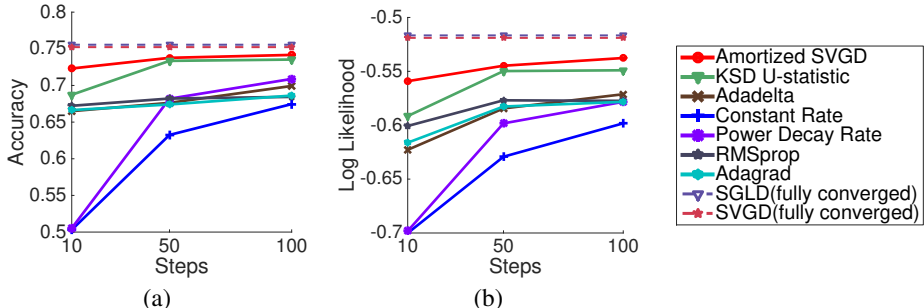

Figure 3: The testing accuracy (a) and testing likelihood (b) when training Langevin inference network with $T \in \{10, 50, 100\}$ layers, respectively. The results reported here are the performance of the final result $z^T$ outputted by the last layer of the network. We find that both amortized SVGD and KSD minimization (with U-statistics) outperform all the hand-designed learning rates. Results averaged on 100 random trails.

**Bayesian Logistic Regression**   We consider Bayesian logistic regression for binary classification using the same setting as Gershman et al. (2012), which assigns the regression weights $w$ with a Gaussian prior $p_0(w|\alpha) = \mathcal{N}(w, \alpha^{-1})$ and $p_0(\alpha) = Gamma(\alpha, 1, 0.01)$. The inference is applied on the posterior of $z = [w, \log \alpha]$. We test this model on the binary Covertype dataset[1] with 581,012 data points and 54 features.

To demonstrate that our estimated learning rate can work well on new datasets never seen by the algorithm. We partition the dataset into mini-datasets of size $50,000$, and use $80\%$ of them for training and $20\%$ for testing. We adapt our amortized SVGD/KSD to train on the whole population of the training mini-datasets by randomly selecting a mini-dataset at each iteration of Algorithm 1, and evaluate the performance of the estimated step sizes on the remaining $20\%$ testing mini-datasets.

Figure 3 reports the testing accuracy and likelihood on the $20\%$ testing mini-datasets when we train the Langevin network with $T = 10, 50, 100$ layers, respectively. We find that our methods outperform all the hand-designed learning rates, and allow us to get performance closer to the fully converged SGLD and SVGD with a small number $T$ of iterations.

Figure 4 shows the testing accuracy and testing likelihood of all the intermediate results when training Langevin network with $T = 100$ layers. It is interesting to observe that amortized SVGD and KSD learn rather different behavior: KSD tends to increase the performance quickly at the first few iterations but saturate quickly, while amortized SVGD tends to increase slowly in the beginning and boost the performance quickly in the last few iterations. Note that both algorithms are set up to optimize the performance of the last layers, while need to decide how to make progress on the intermediate layers to achieve the best final performance.

## 5.2   GENERAL LANGEVIN INFERENCE NETWORK

We further test our algorithm with general Langevin inference network. We firstly construct one single layer general Langevin network to approach the posterior of Bayesian logistic regression parameters and we can achieve $74.58\%$ average accuracy and $-0.5216$ average testing log-likelihood in 100 repeat experiments. This result proves the proposed general Langevin Inference Network is quite competitive and worth to explore. Moreover, we use it as a black-box sampler to approach more complicate Gaussian Mixture distributions.

**Gaussian Mixture**   We consider 10 components Gaussian Mixture Models with mean and covariance matrix of each component randomly drawn from a uniform distribution, and we test our methods on different dimensions models.

We construct 6 layers of general Langevin networks as a black-box sampler, and our proposed two methods to train the black-box sampler to approximate the target distribution. Figure 5 shows our

---

[1]https://www.csie.ntu.edu.tw/~cjlin/libsvmtools/datasets/binary.html

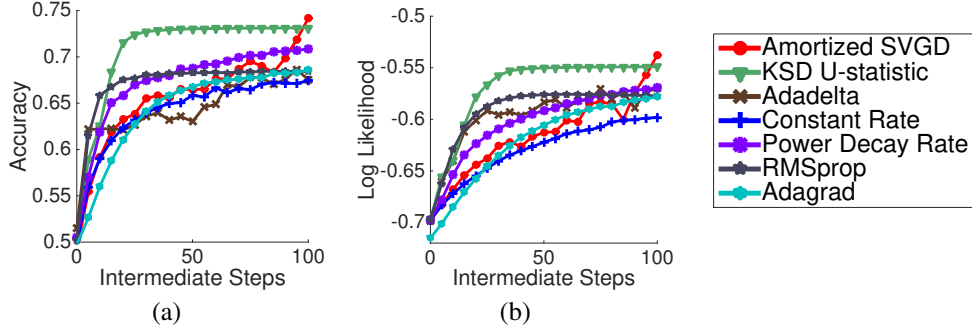

Figure 4: The testing accuracy (a) and testing likelihood (b) of the outputs of the intermediate layers when training the Langevin network with $T = 100$ layers. Note that both amortized SVGD and KSD minimization target to optimize the performance of the last layer, but need to optimize the progress of the intermediate steps in order to achieve the best final results.

results on 50 dimension Gaussian Mixture case and figure 6 shows results of different dimensions of Gaussian Mixture. From the figures we can know that our proposed sampling structure is quite competive comparing with NUT sampler(Hoffman & Gelman, 2014), and these two variational inference methods can both train a good black-box sampler.

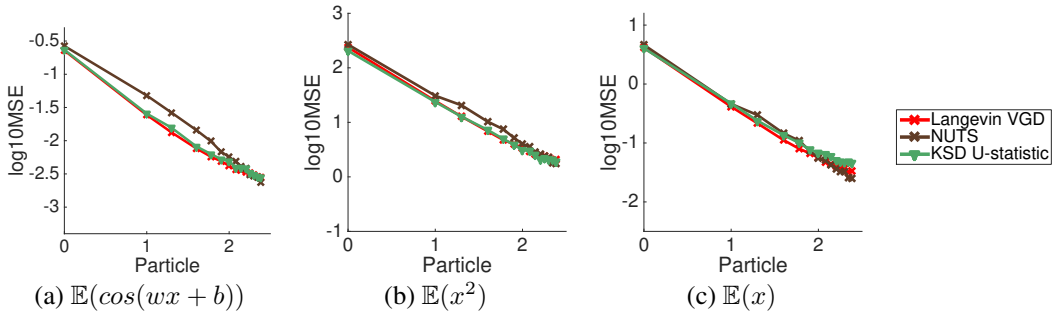

Figure 5: Comparation between our methods and NUTS on 50 dimension Gaussian Mixture. (a)-(c) show the mean square errors when using different number particles to estimate expectation $\mathbb{E}(h(x))$ for $h(x) = x$, $x^2$, and $cos(x + b)$; for $cos(\omega x + b)$, we random draw $\omega \sim \mathcal{N}(0,1)$ and $b \sim$ Uniform$([0, 2\pi])$ and report the average MSE over 10 random draws of and b.

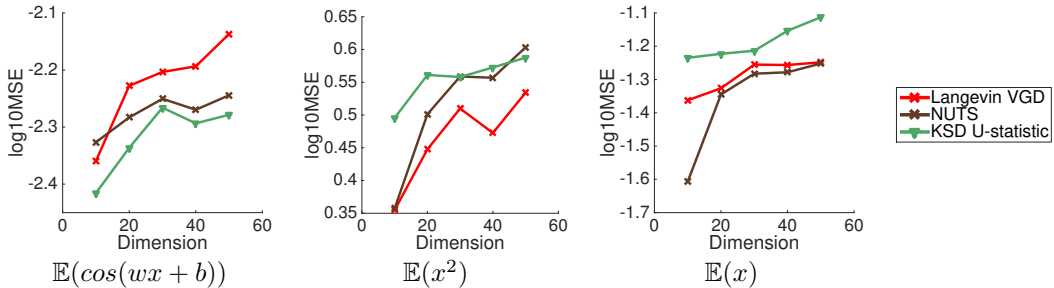

Figure 6: Comparation between our methods and NUTS For different dimension Gaussian Mixture. (a)-(c) show the mean square errors when using different number particles to estimate expectation $\mathbb{E}(h(x))$ for $h(x) = x$, $x^2$, and $cos(x + b)$; for $cos(\omega x + b)$, we random draw $\omega \sim \mathcal{N}(0,1)$ and $b \sim$ Uniform$([0, 2\pi])$ and report the average MSE over 10 random draws of and b.

# 6 CONCLUSION

We consider two methods for wild variational inference that allows us to train general inference networks with intractable density functions, and apply it to adaptively estimate step sizes of stochastic gradient Langevin dynamics. More studies are needed to develop better methods, more applications and theoretical understandings for wild variational inference, and we hope that the two methods we discussed in the paper can motivate more ideas and studies in the field.

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
