# Peer review of "Two Methods for Wild Variational Inference"

_ICLR 2017 — rejected_

[Official Review · AnonReviewer2 · rating 3 · confidence 4 · 15 Dec 2016]
**An unclearly written paper, which is not focused, not clearly original and which lacks a rigorous and critical evaluation.**

The authors propose methods for wild variational inference, in which the
variational approximating distribution may not have a directly accessible
density function. Their approach is based on the Stain's operator, which acts
on a given function and returns a zero mean function with respect to a given
density function which may not be normalized.

Quality:

The derviations seem to be technically sound. However, my impression is that
the authors are not very careful and honest at evaluating both the strengths
and weaknesses of the proposed work. How does the method perform in cases in
which the distribution to be approximated is high dimensional? The logistic
regression problem considered only has 54 dimensions. How would this method
perform in a neural network in which the number of weights is goint to be way
much larger? The logistic regression model is rather simple and its posterior
will be likely to be close to Gaussian. How would the method perform in more
complicated posteriors such as the ones of Bayesia neural networks?

Clarity:

The paper is not clearly written. I found it very really hard to follow and not
focused. The authors describe way too many methods: 1) Stein's variational
gradient descent (SVGD), 2) Amortized SVGD, 3) Kernelized Stein discrepancy
(KSD), 4) Lavengin inference network, not to mention the introduction to
Stein's discrepancy. I found very difficult to indentify the clear
contributions of the paper with so many different techniques.

Originality:

It is not clear how original the proposed contributions are. The first of the
proposed methods is also discussed in

Wang, Dilin and Liu, Qiang. Learning to draw samples: With application to
amortized mle for generative adversarial learning. Submitted to ICLR 2017, 2016

How does this work differ from that one?

Significance:

It is very hard to evaluate the importance of proposed methods. The authors
only report results on a 1d toy problem with a mixture of Gaussians and on a
logistic regression model with dimension 54. In both cases the distributions to
be approximated are very simple and of low dimension. In the regression case
the posterior is also likely to be close to Gaussian and therefore not clear
what advances the proposed method would provide with respect to other more
simple approaches. The authors do not compare with simple variational
approaches based on Gaussian approximations.

[Official Review · AnonReviewer1 · rating 3 · confidence 4 · 16 Dec 2016 (modified: 17 Dec 2016)]
**Review: Two Methods for Wild Variational Inference**

The authors propose two variational methods based on the theme of posterior approximations which may not have a tractable density. The first is from another ICLR submission on "amortized SVGD" (Wang and Liu, 2016), where here the innovation is in using SGLD as the inference network. The second is from a NIPS paper (Ranganath et al., 2016) on minimizing the Stein divergence with a parametric approximating family, where here the innovation is in defining their test functions to be an RKHS, obtaining an analytic solution to the inner optimization problem.

The methodology is incremental. Everything up to Section 3.2 is essentially motivation, background, or related work. The notion of a "wild variational approximation" was already defined in Ranganath et al. (2016), termed a "variational program". It would be useful for the authors to comment on the difference, if any.

Section 3.2 is at first interesting because it analytically solves the maximum problem that is faced in Ranganath et al. (2016). However, this requires use of a kernel which will certainly not scale in high dimensions, so it is then equivalent in practice to having chosen a very simple test function family. To properly scale to high dimensions would require a deeper kernel and also learning its parameters; this is not any easier than parameterizing the test function family as a neural network to begin with, which Ranganath et al. (2016) do.

Section 4 introduces a Langevin inference network, which essentially chooses the variational approximation as an evolving sequence of Markov transition operators as in Salimans et al. (2015). I had trouble understanding this for a while because I could not understand what they mean by inference network. None of it is amortized in the usual inference network sense, which is that the parameters are given by the output of a neural network. Here, the authors simple define global parameters of the SGLD chain which are used across all the latent variables (which is strictly worse?). (What then makes it an "inference network"?) Is this not the variational approximation used in Salimans et al. (2015), but using a different objective to train it?

The experiments are limited, on a toy mixture of Gaussians posterior and Bayesian logistic regression. None of this addresses the problems one might suspect on high-dimensional and real data, such as the lack of scalability for the kernel, the comparison to Salimans et al. (2015) for the Langevin variational approximation, and any note of runtime or difficulty of training.

Minor comments

+ It's not clear if the authors understood previous work on expressive variational families or inference networks. For example, they argue Rezende & Mohamed, 2015b; Tran et al., 2015; Ranganath et al., 2015 require handcrafted inference networks. However, all of them assume use of any neural network for amortized inference. None of them even require an inference network. Perhaps the authors mean handcrafted posterior approximations, which to some extent is true; however, the three mentioned are all algorithmic in nature: in Rezende & Mohamed (2015), the main decision choice is the flow length; Tran et al. (2015), the size of the variational data; Ranganath et al. (2015), the flow length on the auxiliary variable space. Each works well on different problems, but this is also true of variational objectives which admit intractable q (as the latter two consider, as does Salimans et al. (2015)). The paper's motivation could be better explained, and perhaps the authors could be clearer on what they mean by inference network.
+ I also recommend the authors not term a variational inference method based on the class of approximating family. While black box variational inference in Ranganath et al. (2014) assumes a mean-field family, the term itself has been used in the literature to mean any variational method that imposes few constraints on the model class.

[Official Review · AnonReviewer3 · rating 3 · confidence 4 · 17 Dec 2016]
**Official Review**

The paper proposes two methods for what is called wild variational inference. 
The goal is to obtain samples from the variational approximate distribution q 
without requiring to evaluate the density q(z) by which it becomes possible to 
consider more flexible family of distributions. The authors apply the proposed 
method to the problem of optimizing the hyperparamter of the SGLD sampler. 
The experiments are performed on a 1-d mixture of gaussian distribution and 
Bayesian logistic regression tasks. 

The key contribution seems to connect the previous findings in SVGD and KSD 
to the concept of inference networks, and to use them for hyperparameter 
optimization of SGLD. This can not only be considered as a rather simple 
connection/extension, but also the toyish experiments are not enough to convince 
readers on the significance of the proposed model. Particularly, I'm wondering 
how the particle based methods can deal with the multimodality (not the simple
1d gaussian mixture case) in general. Also, the method seems still to require to evaluate
the true gradient of the target distribution (e.g., the posterior distribution) for 
each z ~ q. This seems to be a computational problem for large dataset settings. 
In the experiments, the authors compare the methods for the same number of 
update steps. But, considering the light computation of SGLD per update, I think 
SGLD can make much more updates per unit time than the proposed methods, 
particularly for large datasets. The Bayesian logistic regression on 54 dimensions
seems also a quite simple experiment, considering that its posterior is close to 
a Gaussian distribution. Also, including Hamiltonian Monte Carlo (HMC) with 
automatic hyperparameter tuning mechanism (like, no u-turn sampler) would be
interesting.

The paper is written very unclearly. Especially, it is not clear what is the exact
contributions of the paper compared to the other previous works including the
authors' works. The main message is quite simple but most of the pages are 
spent to explain previous works. 

Overall, I'd like to suggest to have more significant high-dimension, large scale 
experiments, and to improve the writing.

[Author Response · Yihao Feng · 24 Jan 2017]
**Thank you for your review and comments**

We highly appreciate the time and feedback from all the reviewers, all of which we will take into serious consideration in our revision. We agree that this paper needs to be re-structured significantly. We will significantly clarify the presentation and its relation with existing reference and also strengthen the empirical evaluation. Currently, we have shown that our algorithms work well with multi-modal distributions such as GMM and RBM, and is promising when applied on VAE.

[Final Decision · Program Chairs · 06 Feb 2017]
**ICLR committee final decision**

This paper is both time and topical in that it forms part of the growing and important literature of ways of representing approximate posterior distributions for variational inference that do need need a known or tractable density. The reviewers have identified a number of areas that when addressed will improve the paper greatly. These include: more clearer and structured introduction of methods, with the aim of highlighting how this work adds to the existing literature; careful explanation of the term wild variational inference, especially in context of alternative terms and more on relation to existing work; experiments on higher-dimensional data, much greater than 54 dimensions, to help better understand the advantages and disadvantages of this approach. It if for these reasons that the paper at this point is not yet ready for acceptance at the conference.